# Associations of cerebral oxygenation with hemoglobin levels evaluated by near-infrared spectroscopy in hemodialysis patients

**Susumu Ookawara**[1][☯]*, **Kiyonori Ito**[1][☯], **Yusuke Sasabuchi**[2], **Hideyuki Hayasaka**[3], **Masaya Kofuji**[3], **Takayuki Uchida**[3], **Keita Horigome**[4], **Sojiro Imai**[5], **Toshiko Akikawa**[4], **Noriko Wada**[5], **Satoshi Kiryu**[5], **Satoru Imada**[4], **Mitsutoshi Shindo**[1], **Haruhisa Miyazawa**[1], **Keiji Hirai**[1], **Yasushi Onishi**[6], **Hirofumi Shimoyama**[7], **Akihisa Watanabe**[8], **Kaoru Tabei**[6], **Yoshiyuki Morishita**[1]

1 Division of Nephrology, First Department of Integrated Medicine, Saitama Medical Center, Jichi Medical University, Saitama, Japan, 2 Data Science Center, Jichi Medical University, Tochigi, Japan, 3 Department of Clinical Engineering, Saitama Medical Center, Jichi Medical University, Saitama, Japan, 4 Division of Hemodialysis, Yuai Nakagawa Clinic, Hakuyukai Medical Corporation, Saitama, Japan, 5 Department of Dialysis, Minami-uonuma City Hospital, Niigata, Japan, 6 Department of Internal Medicine, Minami-uonuma City Hospital, Niigata, Japan, 7 Division of Nephrology, Yuai Nisshin Clinic, Hakuyukai Medical Corporation, Saitama, Japan, 8 Division of Nephrology, Yuai Nakagawa Clinic, Hakuyukai Medical Corporation, Saitama, Japan

☯ These authors contributed equally to this work.
* su-ooka@hb.tp1.jp

**Data Availability Statement:** All relevant data are within the paper.

## Abstract

Hemoglobin (Hb) is associated with cerebral oxygenation status owing to its important role of carrying oxygen to systemic tissues. However, data concerning the associations between Hb levels and cerebral regional oxygen saturation ($rSO_2$) of hemodialysis (HD) patients is limited. We aimed to identify these associations to consider a target Hb level for renal anemia management. This study included 375 HD patients. Cerebral $rSO_2$ before HD was monitored using the INVOS 5100c oxygen saturation monitor. Multivariable linear regression analysis showed that cerebral $rSO_2$ was independently associated with natural logarithm (Ln)-HD duration (standardized coefficient: -0.36), mean blood pressure (standardized coefficient: 0.13), pH (standardized coefficient: -0.10), serum albumin (standardized coefficient: 0.14), presence of diabetes mellitus (standardized coefficient: -0.20), and Hb level (standardized coefficient: 0.29). Furthermore, a generalized linear model with restricted cubic spline function was used to investigate the non-linear association between cerebral $rSO_2$ and Hb levels. In the multivariable analysis for the adjustment with Ln-HD duration, mean blood pressure, pH, serum albumin, and presence of diabetes mellitus, a linear relationship was demonstrated between the two variables ($p$ for linearity = 0.79). Hb levels revealed the positive and significant association with cerebral $rSO_2$ in this study. Moreover, the relationship between cerebral $rSO_2$ and Hb level was proven to be linear. Therefore, the target Hb level in renal anemia management would be considered to be the upper limits for the appropriate management of renal anemia by previous guidelines and position statement from the viewpoint of maintaining cerebral oxygenation in HD patients.

**Funding:** This work was supported by grants from the Japanese Association of Dialysis Physicians (JADP Grant 2017-9), The Kidney Foundation, Japan (JKFB 18-7), and JSPS KAKENHI under Grant No. JP20K11534 to SO. Additionally, Hakuyukai Medical Corporation provided support in the form of salaries for authors KH, TA, S. Imada, HS, and AW. The specific roles of these authors are articulated in the 'author contributions' section. The funders had no role in study design, data collection and analysis, decision to publish, or manuscript preparation.

**Competing interests:** Hakuyukai Medical Corporation provided support in the form of salaries for authors KH, TA, S. Imada, HS, and AW. There are no patents, products in development or marketed products associated with this research to declare. This does not alter our adherence to PLOS ONE policies on sharing data and materials.

## Introduction

Renal anemia is a common complication in patients with advanced chronic kidney disease (CKD), including hemodialysis (HD) patients. Under a severe anemic status caused by the progression of renal anemia, it is largely responsible for the significant detrimental effects of CKD including decreases in exercise capacity, immune response, cognitive function, and nutrition, and increases in depression, cardiac dysfunction, morbidity, and mortality [1]. The Kidney Disease Improving Global Outcomes Guidelines [2], the European Renal Best Practice position statement [3], and Guidelines for Renal Anemia edited by the Japanese Society for Dialysis Therapy [4], have been proposed for the appropriate management of renal anemia, and are clinically important to standardize renal anemia management. However, these guidelines and position statement were determined by randomized controlled trials and clinical trials such as comparison of the quality of life and differences in mortality [5–11]. Therefore, there is a possibility that they do not accurately reflect physiological changes in response to Hb level changes.

Recently, cerebral oxygenation evaluation has attracted attention in the field of CKD and HD therapy because of its influence on clinical conditions. In particular, cerebral regional oxygen saturation ($rSO_2$) using near-infrared spectroscopy (NIRS) has been easily, noninvasively, and automatically measured in CKD patients, including those undergoing HD [12–15], and might be suggested the association with cognitive function [16,17]. Regarding the association between cerebral $rSO_2$ and clinical parameters, the Hb levels did not affect cerebral $rSO_2$ in HD patients with well-maintained Hb levels [13,14], whereas a significant increase in cerebral $rSO_2$ was observed in response to the Hb increase after blood transfusion in severely anemic HD patients [15]. This may suggest the existence of a transitional range in the association between cerebral $rSO_2$ and Hb level. On the other hand, several clinical parameters were reported to influence the cerebral $rSO_2$ values [14]; therefore, cerebral $rSO_2$ changes were not determined by the Hb levels alone. To date, only few reports have investigated the relationship between cerebral $rSO_2$ and Hb level across a wide range of Hb levels. Therefore, this study aimed to investigate the association of Hb levels with cerebral $rSO_2$ values under the adjustment with other significant clinical parameters affecting cerebral oxygenation, and determine the target Hb level from the viewpoint of maintaining and improving cerebral oxygenation in HD patients.

## Materials and methods

### Patients

In this study, patients were included upon meeting the following criteria: (i) age > 20 years; and (ii) end-stage renal disease managed with intermittent HD or hemodiafiltration (HDF). The exclusion criteria were: (i) coexisting major disease including congestive heart failure or neurological disorders, and (ii) a history of cerebrovascular disease.

A total of 452 patients met the inclusion criteria and were enrolled from July 1, 2013 to August 31, 2017. However, 77 patients were excluded from analysis because of missing data. Therefore, 375 patients were finally included in this study (257 men, 118 women; mean age, 68 ± 11 years; mean HD duration, 5.4 ± 6.7 years; Table 1). Each patient underwent HD or HDF 2–3 times per week for 3–4 h per session. The causes of chronic renal failure included type 2 diabetes mellitus (DM) (164 patients), chronic glomerulonephritis (85 patients), nephrosclerosis (58 patients), and others (68 patients). In this study, 310 and 65 patients received HD and HDF, respectively.

**Table 1. Hemodialysis patient characteristics and correlations between cerebral rSO$_2$ and clinical parameters.**

| | Mean ± SD | Simple linear regression vs. cerebral rSO$_2$ values | |
|---|---|---|---|
| | | r | p value |
| Patient characteristics | | | |
| Number of patients, n (men/women) | 375 (257/118) | | |
| Cerebral rSO$_2$ (%) | 51.0 ± 9.5 | | |
| Age (years) | 68 ± 11 | -0.03 | 0.61 |
| HD duration (years), median (interquartile range) | 2.5 (0.2–8.4) | -0.27 | < 0.001* |
| Causes of chronic renal failure | | | |
| Diabetes mellitus, n (%) | 164 (44) | | |
| Chronic glomerulonephritis, n (%) | 85 (23) | | |
| Nephrosclerosis, n (%) | 58 (15) | | |
| Others, n (%) | 68 (18) | | |
| Body weight (kg) | 59.9 ± 12.9 | 0.06 | 0.21 |
| Body Mass Index (kg/m$^2$) | 22.9 ± 4.1 | 0.08 | 0.12 |
| Interdialytic weight gain (kg) | 1.9 ± 1.0 | 0.04 | 0.49 |
| Interdialytic weight gain/dry weight (%) | 3.4 ± 1.8 | 0.07 | 0.19 |
| Mean blood pressure (mmHg) | 99 ± 15 | 0.22 | < 0.001* |
| Heart rate (/min) | 74 ± 14 | -0.07 | 0.16 |
| Use of erythropoiesis-stimulating agent, n (%) | 337 (90) | | |
| Erythropoiesis-stimulating agent (IU/week) | 6105 ± 4616 | -0.14 | 0.006* |
| Iron supplementation, n (%) | 67 (18) | | |
| Laboratory findings | | | |
| pH | 7.37 ± 0.04 | -0.24 | < 0.001* |
| pO$_2$ (mmHg) | 85 ± 16 | 0.04 | 0.40 |
| Sat O$_2$ (%) | 95 ± 5 | 0.01 | 0.80 |
| Hb (g/dL) | 10.4 ± 1.5 | 0.32 | < 0.001* |
| Serum iron (μg/dL) | 56 ± 29 | 0.22 | < 0.001* |
| Total iron binding capacity (μg/dL) | 229 ± 51 | 0.15 | <0.01 |
| Transferrin saturation (%) | 25 ± 14 | 0.12 | 0.02* |
| Serum ferritin (ng/mL) | 147 ± 219 | -0.05 | 0.31 |
| Blood urea nitrogen (mg/dL) | 58 ± 17 | 0.19 | < 0.001* |
| Serum creatinine (mg/dL) | 9.2 ± 2.7 | 0.20 | < 0.001* |
| Total protein (g/dL) | 5.9 ± 1.7 | 0.18 | < 0.001* |
| Serum albumin (g/dL) | 3.3 ± 0.6 | 0.36 | < 0.001* |

Values are shown as mean ± standard deviation (SD) except where otherwise indicated.

*Statistically significant.

Abbreviations: rSO$_2$, regional oxygen saturation; HD, hemodialysis; Hb, hemoglobin.

## Ethical approval

All participants provided written informed consent to participate. The study was approved by the Institutional Review Board of the Saitama Medical Center, Jichi Medical University, Japan (RIN 15–104), Minami-uonuma City Hospital, Japan (H29-11), and Yuai Nakagawa Clinic, Japan (28005) and conformed to the provisions of the Declaration of Helsinki (as revised in Tokyo in 2004).

**Patient baseline characteristics and clinical laboratory measurements.** Patients' baseline characteristics and other relevant data were collected from their medical charts. The primary disease underlying the requirement for dialysis and the coexistence of cardiovascular disease were extracted from their medical records.

Blood pressure and heart rate were measured with the patients in the supine position before HD or HDF. Blood samples were obtained at an ambient temperature from the HD access point, including arteriovenous fistulas, arteriovenous grafts, and HD catheters, of each patient before HD and peripheral blood count including Hb level and biochemical parameters were measured.

## Cerebral oxygenation monitoring

Cerebral $rSO_2$, cerebral oxygenation marker, was monitored using an INVOS 5100c saturation monitor (Covidien Japan, Tokyo, Japan). This instrument used a light-emitting diode that transmitted near-infrared light at two wavelengths (735 and 810 nm) and two silicon photodiodes that acted as light detectors to measure oxygenated and deoxygenated Hb. The ratio of oxygenated Hb to the total Hb (oxygenated Hb + deoxygenated Hb) signal strength was calculated and the corresponding percentage was read as a single numerical value that represented the $rSO_2$ [18,19]. All data obtained by this instrument were immediately and automatically stored in sequence. The inter-observer variance of this instrument (i.e., the reproducibility of the $rSO_2$ measurement) was acceptable, as previously reported [20–22]. Therefore, $rSO_2$ was considered reliable for estimating actual cerebral oxygenation levels. Furthermore, the light paths leading from the emitter to the different detectors shared a common part: the 30 mm detector assessed superficial tissue, while the 40 mm detector was used to assess deep tissue. By analyzing the differential signals collected by the two detectors, the current data for $rSO_2$ and total Hb signal strength values were obtained in deep tissue at 20–30 mm from the body surface [23,24]. These measurements were performed at 6-s intervals.

In this study, the cerebral $rSO_2$ measurement was performed at first HD session per weekly HD therapy in 241 patients, second HD session in 85 patients, and third HD session in 49 patients. Prior to HD, the recruited patients rested in the supine position for at least 10 min in order to reduce the influence of postural changes on $rSO_2$. An $rSO_2$ measurement sensor was attached to the patient's forehead for measurement in the resting state. Thereafter, $rSO_2$ was measured for 5 min before HD, and the mean $rSO_2$ was calculated as a measure of cerebral oxygenation in each patient.

## Statistics

Data were expressed as mean ± standard deviation or median and interquartile range. Correlations between the two groups were evaluated using the Pearson's correlation coefficient and linear regression analysis. The Student's t-test for non-paired values was used for comparing 2 groups. Variables with a p value below 0.05 in a simple linear regression analysis and plausible confounding factors were included in the multivariable linear regression analysis to identify factors affecting cerebral $rSO_2$ in HD patients. Furthermore, a generalized linear model with restricted cubic spline function was used to investigate the non-linear association between cerebral $rSO_2$ and Hb levels. HD duration was transformed using the natural logarithm (Ln) in the regression analyses because of the skewed distribution. All analyses were performed using the IBM SPSS Statistics for Windows, version 26.0 (IBM, Armonk, NY, USA). Additionally, R V.3.4.1 (The R Foundation, Vienna, Austria) and 'rms' package was used for a restricted cubic spline function. $P < 0.05$ was considered statistically significant.

## Results

Table 1 shows patients' characteristics and correlations between cerebral $rSO_2$ and clinical parameters. Cerebral $rSO_2$ showed significant positive correlations with mean blood pressure, Hb level, serum iron, total iron binding capacity, transferrin saturation, blood urea nitrogen,

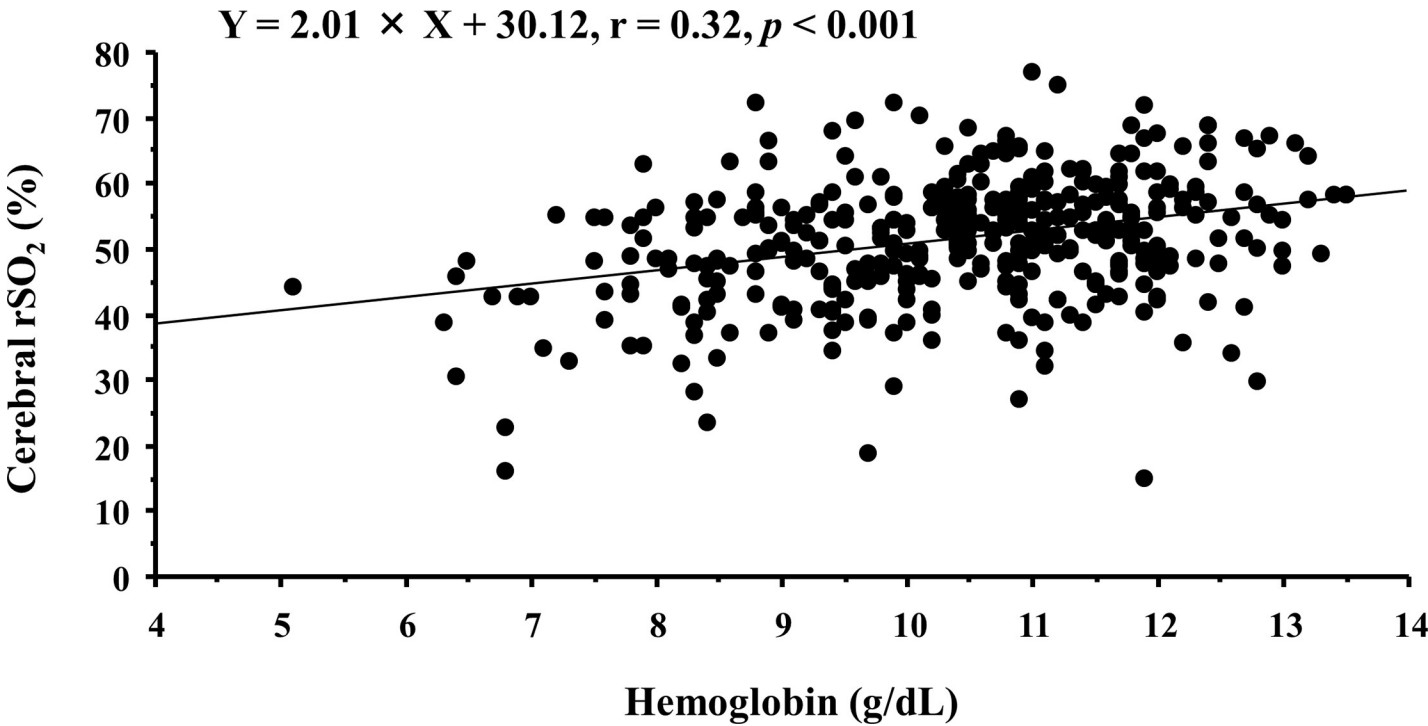

**Fig 1. Simple linear correlation between cerebral rSO₂ and Hb levels in all patients.** Abbreviations: rSO$_2$, regional oxygen saturation; Hb, hemoglobin.

serum creatinine, total protein, and serum albumin concentration, in addition to the negative correlations with Ln-HD duration, dose of erythropoiesis-stimulating agent (ESA), and pH. As shown in Fig 1, there is a significant positive correlation between cerebral rSO$_2$ and Hb level (cerebral rSO$_2$ = 2.01 × Hb level + 30.12, r = 0.32, $p < 0.001$). Furthermore, cerebral rSO$_2$ was significantly lower in HD patients with DM than in those without DM (48.6 ± 9.3% vs. 52.9 ± 9.3%, $p < 0.001$, Fig 2).

Results of the multivariable linear regression analysis are presented in Table 2. Ln-HD duration, mean blood pressure, dose of ESA, pH, Hb level, transferrin saturation, serum creatinine, and serum albumin as variables with p values below 0.05 and presence of DM as a confounding factor were included in the multivariable linear regression analysis. As shown in Table 2, cerebral rSO$_2$ was independently associated with Ln-HD duration (standardized coefficient: -0.36), mean blood pressure (standardized coefficient: 0.13), pH (standardized coefficient: -0.10), serum albumin (standardized coefficient: 0.14), presence of DM (standardized coefficient: -0.20), and Hb level (standardized coefficient: 0.29).

Furthermore, we conducted a regression analysis using a generalized linear model with restricted cubic spline function to test a non-linear relationship between cerebral rSO$_2$ and Hb levels. We adjusted for Ln-HD duration, mean blood pressure, pH, serum albumin, and presence of DM, which were independently and significantly associated with cerebral rSO$_2$ in the multivariable linear regression analysis, and the cerebral rSO$_2$ and Hb levels demonstrated a linear relationship ($p$ for linearity = 0.79, Fig 3).

## Discussion

Hb plays an important role in carrying oxygen to systemic tissues, including the brain, and therefore, systemic oxygenation is believed to be associated with Hb levels. It has been

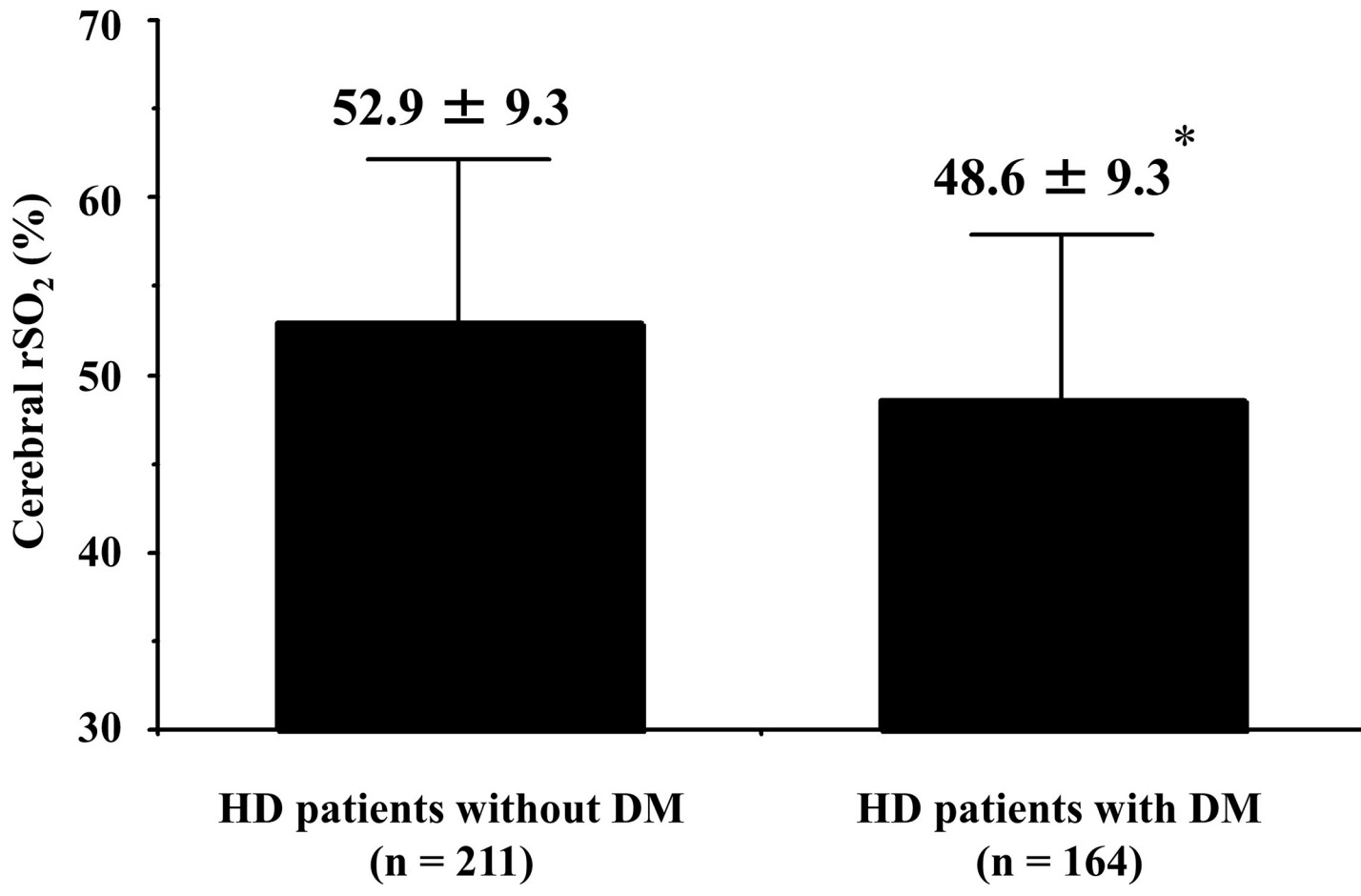

**Fig 2. Comparison of cerebral rSO$_2$ in HD patients with and without diabetes mellitus.** $^*$ $p < 0.001$ vs. HD patients without DM. Abbreviations: rSO$_2$, regional oxygen saturation; HD, hemodialysis; DM, diabetes mellitus.

previously reported that there was a strong correlation between Hb levels and cerebral rSO$_2$ values in patients without CKD with mean Hb levels under 12.9 ± 2.0 g/dL [25]. Furthermore, cerebral oxygen saturation was lower in patients with anemia than those without anemia (with

**Table 2. Multivariable linear regression analysis: Independent factors of cerebral rSO$_2$ in HD patients.**

| Variables | Coefficient | Standardized coefficient | $p$ |
|---|---|---|---|
| Ln-HD duration | -1.71 | -0.36 | < 0.001$^*$ |
| Mean blood pressure | 0.08 | 0.13 | 0.005$^*$ |
| Erythropoiesis-stimulating agent | | -0.04 | 0.437 |
| pH | -22.9 | -0.10 | 0.024$^*$ |
| Hb | 1.83 | 0.29 | < 0.001$^*$ |
| Serum creatinine | | 0.05 | 0.360 |
| Serum albumin | 2.40 | 0.14 | 0.007$^*$ |
| Transferrin saturation | | 0.06 | 0.144 |
| Presence of diabetes mellitus | -3.89 | -0.20 | < 0.001$^*$ |

$^*$Statistically significant.

Abbreviations: HD, hemodialysis; Hb, hemoglobin.

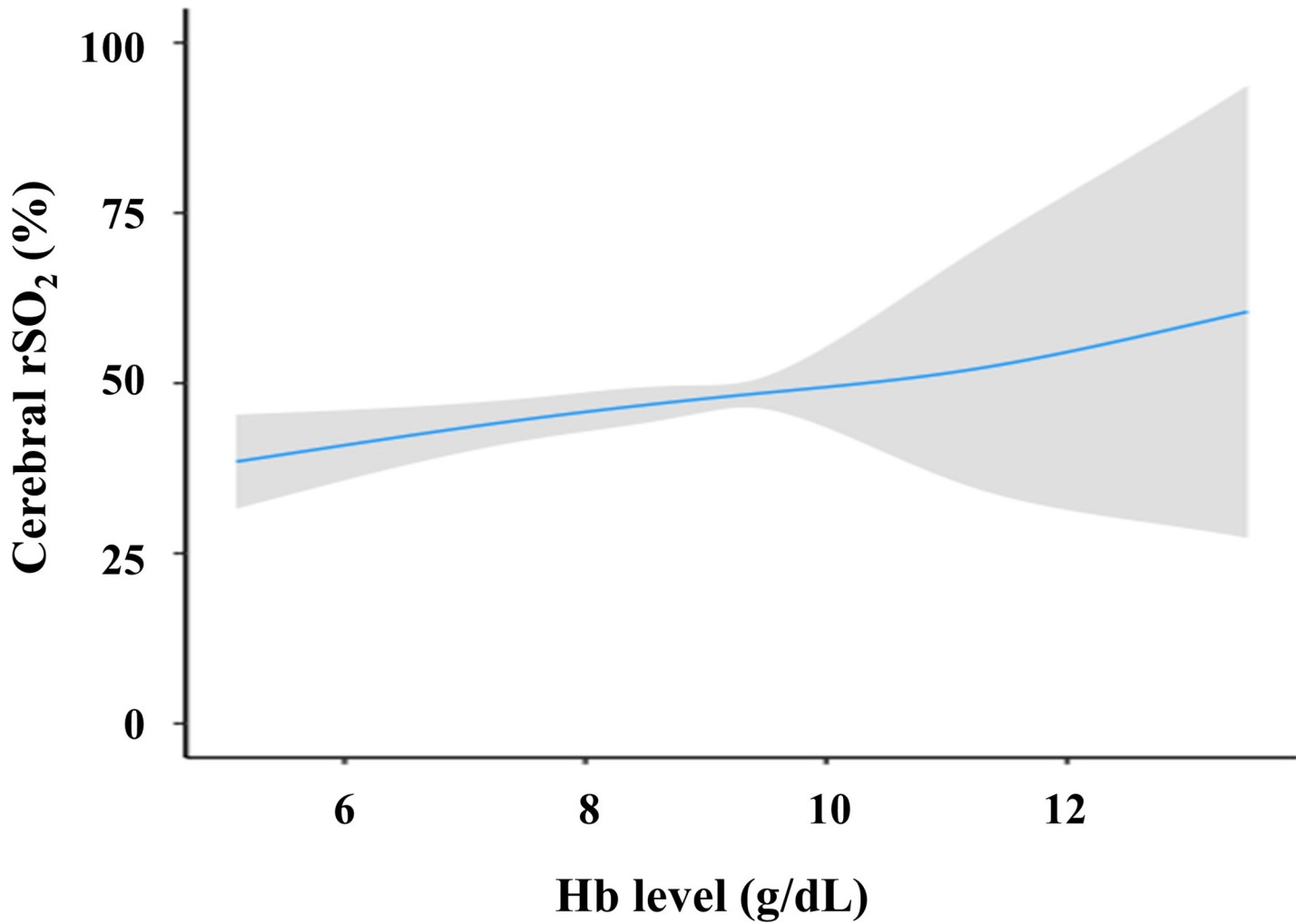

**Fig 3. Adjusted association of Hb levels with cerebral rSO$_2$ values.** In the generalized linear model, we adjusted for Ln-HD duration, mean blood pressure, pH, serum albumin, and presence of DM. The solid line represents the point of estimate and the gray zone represents the 95% confidential interval. Abbreviations: rSO$_2$, regional oxygen saturation; Hb, hemoglobin; Ln-HD duration, natural logarithm of hemodialysis duration; DM, diabetes mellitus.

anemia: Hb level 8.7 ± 2.3 g/dL, cerebral rSO$_2$: 50 ± 11%; without anemia: Hb level 12.3 ± 4.2 g/dL, cerebral rSO$_2$: 66 ± 8%) [26]. On the other hand, cerebral rSO$_2$ values were not significantly associated with Hb levels in HD patients with well-maintained Hb levels despite the significant association with pH, HD duration, serum albumin concentration, and presence of DM [14]. However, in severely anemic HD patients, cerebral rSO$_2$ values significantly increased in response to the Hb increase from intradialytic blood transfusions [15]. In addition, changes in cerebral rSO$_2$ induced by intradialytic blood transfusion were independently associated with the ratio of Hb levels before and after blood transfusion [27]. Based on these results, a transitional range in the association between cerebral rSO$_2$ and Hb levels would be suspected, according to which the association would be positive and significant in patients with anemic status, whereas the association would disappear in patients with well-maintained Hb levels. Therefore, we collected large number of HD patients with widely distributed Hb levels compared with previous studies [13,14], and investigated to identify the clinical parameters, including Hb level, significantly associated with cerebral rSO$_2$ values. As a result, cerebral rSO$_2$ was significantly associated with Hb levels in addition to the HD duration, mean blood

pressure, pH, serum albumin, and presence of DM. Furthermore, after the adjustment for Ln-HD duration, mean blood pressure, pH, serum albumin, and presence of DM using multivariable analysis, linear association between cerebral rSO$_2$ and Hb levels was proven in this study.

To prevent anemia-associated complications in HD patients, appropriate Hb ranges for renal anemia management have been proposed in several sets of guidelines and position statement. The Kidney Disease Improving Global Outcomes Guidelines proposed that an acceptable range for Hb in renal anemia management is 9.5–11.5 g/dL and not exceeding 11.5 g/dL [2]. The European Renal Best Practice position statement recommended that Hb values should not be allowed to routinely fall below 10 g/dL, and that the start of ESA therapy could be considered at higher Hb values but not exceeding 12 g/dL [3]. Furthermore, Guidelines for Renal Anemia edited by the Japanese Society for Dialysis Therapy recommended that the target Hb levels to be maintained is 10–12 g/dL [4], based on the results of randomized controlled trials and clinical trials [5–11]. Although these guidelines and position statement included some differences in the target range for Hb levels, these Hb ranges were determined mainly on the basis of the mortality and quality of life. This study focused on the association between Hb levels and cerebral oxygenation and found a linear relationship between the two variables. Therefore, from the viewpoint of maintaining cerebral oxygenation based on this study in HD patients, the target Hb level in renal anemia management would be considered to be 11.5 g/dL, not exceeding 11.5 g/dL, or 12.0 g/dL, not exceeding 12.0 g/dL, which were the upper limits of a target Hb level proposed for the appropriate management of renal anemia by previous guidelines and position statement [2–4].

In this study, mean blood pressure showed positive and significant association with cerebral rSO$_2$ in the multivariable linear regression analysis, which was different from the previous report [14]. Cerebral blood flow is held steady between mean blood pressure 60 and 150 mmHg, which was known as cerebral autoregulation to protect the cerebral tissue from fluctuating systemic blood pressure [28]. Furthermore, cerebral rSO$_2$ was reported to become linearly dependent on BP, in particular, below a mean blood pressure of 60 mmHg [29]. Additionally, in HD patients, impaired cerebral autoregulation due to vascular damages and HD-associated risk factors for hemodynamic instability during HD are likely to lead to the cerebral hypoperfusion [30]. Actually, cerebral perfusion pressure, defined as the difference between mean blood pressure and intracranial pressure, depends largely on the mean blood pressure during HD, and a larger ultrafiltration volume and rate, which sometimes lead to the decrease in mean blood pressure, were reported to be associated with lower cerebral blood flow [31]. Based on these reports, mean blood pressure might be associated with changes in oxygen supply and those in cerebral oxygenation through the impaired cerebral autoregulation and hemodynamic stress during HD. However, in this study, cerebral rSO$_2$ measurements were performed only before HD. Therefore, we cannot directly comment on the association between changes in cerebral oxygenation and those in mean blood pressure influenced by the hemodynamic stress during HD.

This study had several limitations which should be noted. First, in this study, cognitive assessment could not be performed because of the limits of medical examination time for each patient. Renal anemia has been reported to be associated with cognitive impairment [32–34]. Thus, it would be expected that maintaining or improving cerebral rSO$_2$ via the effect of the increased Hb levels is associated with changes in cognitive function. However, to date, there were only a few insufficient reports to examine the association between the cerebral oxygenation and cognitive function [16,17]. In addition, there was no difference in Hb levels between patients with normal versus cognitive impairment [16] and Hb levels showed no significant association with cerebral oxygenation in the multivariable linear regression analysis [17]. Consequently, the association between the cerebral rSO$_2$ affected by the Hb levels and cognitive

function remains uncertain. To confirm the effect of cerebral $rSO_2$ affected by the Hb levels to the changes in cognitive function, a randomized controlled trial would be necessary. Second, we could not unify the HD session of the week to evaluate cerebral oxygenation and blood samples in this study. Differences in interdialytic period in each patient may influence the results of blood chemistry and blood gas analysis as well as the increase in interdialytic weight gain; therefore, cerebral $rSO_2$ values before HD might be influenced by interdialytic changes in these parameters. In this study, there were no significant associations between cerebral $rSO_2$ before HD and interdialytic weight gain (kg), and interdialytic weight gain/dry weight (%) in a simple linear regression analysis, whereas we did not evaluate the cerebral $rSO_2$ values at the end of the last HD session. Therefore, we cannot directly comment on the association between interdialytic changes in cerebral oxygenation and those in blood chemistry and blood gas analysis-associated parameters, and the increase in interdialytic weight gain. Third, inflammation shown in the elevated C-reactive protein is an exacerbating factor for renal anemia [35] and may be associated with changes in cerebral oxygenation via the decrease of Hb levels. However, in this study, not all patients included in this study measured C-reactive protein at the same time with cerebral $rSO_2$ measurements. Therefore, we could not examine the influence of inflammation to the cerebral oxygenation. Furthermore, there were 77 excluded patients because of the insufficient application of blood samples in this study, and would be relatively large number to 452 included patients. Finally, this study could not determine the lower limit of a target Hb level because of the linear association between cerebral $rSO_2$ and Hb levels. Therefore, additional studies are needed to confirm the association between Hb level and cerebral oxygenation in the future.

## Conclusion

Hb levels were significantly associated with cerebral $rSO_2$ in the multivariable linear regression analysis. Furthermore, it was noted that cerebral $rSO_2$ and Hb level had a linear relationship. Therefore, the target Hb level in renal anemia management would be considered to be the upper limits for the appropriate management of renal anemia by previous guidelines and position statement from the viewpoint of maintaining cerebral oxygenation in HD patients.

## Acknowledgments

We thank the study participants and the staff of the hospitals' clinical dialysis centers.

## Author Contributions

**Conceptualization:** Susumu Ookawara, Kiyonori Ito, Yoshiyuki Morishita.

**Data curation:** Susumu Ookawara, Kiyonori Ito.

**Formal analysis:** Susumu Ookawara, Kiyonori Ito, Yusuke Sasabuchi.

**Funding acquisition:** Susumu Ookawara.

**Investigation:** Susumu Ookawara, Kiyonori Ito, Hideyuki Hayasaka, Masaya Kofuji, Takayuki Uchida, Keita Horigome, Sojiro Imai, Toshiko Akikawa, Satoshi Kiryu, Satoru Imada, Mitsutoshi Shindo, Haruhisa Miyazawa, Keiji Hirai.

**Methodology:** Susumu Ookawara, Kiyonori Ito.

**Project administration:** Susumu Ookawara, Kiyonori Ito.

**Supervision:** Susumu Ookawara, Kiyonori Ito, Noriko Wada, Yasushi Onishi, Hirofumi Shimoyama, Akihisa Watanabe, Kaoru Tabei, Yoshiyuki Morishita.

**Validation:** Susumu Ookawara, Kiyonori Ito.

**Writing – original draft:** Susumu Ookawara, Kiyonori Ito.

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
