## [Decision Letter · Decision Letter 0]

18 May 2020

PONE-D-20-05681

Association of cerebral oxygenation evaluated by near infrared spectroscopy with hemoglobin levels in patients undergoing hemodialysis

PLOS ONE

Dear Pr Ookawara,

Thank you for submitting your manuscript to PLOS ONE. After careful consideration, we feel that it has merit but does not fully meet PLOS ONE’s publication criteria as it currently stands. Therefore, we invite you to submit a revised version of the manuscript that addresses the points raised during the review process.

The manuscript is of potential interest. However, it is not acceptable for publication in its current form. I believe the manuscript would benefit from providing more detailed both methodology and results description. For that reason I would ask you to provide the following information:

- the authors should clearly report ( both in the abstract and in the discussion) that if maintaining Hb levels above the lower limit would prevent the decrease of cerebral oxygenation, which might contribute to the maintenance of cognitive function, should be tested in a randomized controlled adequately powered trial specifically designed at this aim.

- please provide information about iron supplement and iron parameters

- please specify before which session of the weekly HD routine the rSO2 measurements were performed

- please consider performing additional statistical analyses as suggested by the Reviewers

- please revise the suggested clinical implications

- please acknowledge all the study limitation as pointed out by the Reviewers

We would appreciate receiving your revised manuscript by Jul 02 2020 11:59PM. To enhance the reproducibility of your results, we recommend that if applicable you deposit your laboratory protocols in protocols.io, where a protocol can be assigned its own identifier (DOI) such that it can be cited independently in the future. For instructions see: http://journals.plos.org/plosone/s/submission-guidelines#loc-laboratory-protocols

We look forward to receiving your revised manuscript.

Kind regards,

Justyna Gołębiewska

Academic Editor

PLOS ONE

Journal Requirements:

"This work was supported by a grant from the Japanese Association of Dialysis Physicians (JADP Grant 2017-9) and a grant from The Kidney Foundation, Japan (JKFB 18-7) to SO."

We note that one or more of the authors are employed by a commercial company: Hakuyukai Medical Corporation.

Reviewers' comments:

Reviewer's Responses to Questions

**Comments to the Author**

1. Is the manuscript technically sound, and do the data support the conclusions?

Reviewer #1: Partly

Reviewer #2: Partly

Reviewer #3: Partly

2. Has the statistical analysis been performed appropriately and rigorously? 

Reviewer #1: No

Reviewer #2: No

Reviewer #3: Yes

3. Have the authors made all data underlying the findings in their manuscript fully available?

Reviewer #1: Yes

Reviewer #2: Yes

Reviewer #3: Yes

4. Is the manuscript presented in an intelligible fashion and written in standard English?

Reviewer #1: Yes

Reviewer #2: Yes

Reviewer #3: Yes

5. Review Comments to the Author

Reviewer #1: The paper “Association of cerebral oxygenation evaluated by near infrared spectroscopy with hemoglobin levels in patients undergoing hemodialysis ( HD)” by Susumu Ookawara et al. deals with an interesting and novel topic :” the association between Hb levels and cerebral oxygen saturation ( rSO2 ) of patients undergoing hemodialysis”. This multi-center study included 375 patients undergoing HD and rSO2 before HD was monitored using the INVOS 5100c oxygen saturation monitor.

The authors found a significant positive correlation between Hb levels and cerebral rSO2 in all patients The slope of correlation line and correlation coefficient between Hb levels and cerebral rSO2 were highest at Hb levels below 10.8 g/dL There was no such correlation with Hb levels above 10.8 g/dL.

The conclusions of the authors were that cerebral rSO2 improved in accordance with Hb increase up to 10.8 g/dL, which could be considered the lower limit of a target Hb level in the management of renal anemia in patients undergoing HD.

General Comments

Strictly speaking this was not an observational study considering that the authors planned to measure the cerebral oxygen saturation of patients undergoing hemodialysis.

The authors should clearly report ( both in the abstract and in the discussion session) that if maintaining Hb levels above the lower limit would prevent the decrease of cerebral oxygenation, which might contribute to the maintenance of cognitive function,. should be tested in a randomized controlled adequately powered trial specifically designed at this aim. The authors only reported that cerebral rSO2 showed a significant association with cognitive function in patients with advanced CKD, including those undergoing HD [15,16].Unfortunately association is not a demonstration.

Specific comments

Introduction :

the authors should clarify that only a severe anemia is responsible of the severe side effects reported here

reference 3 should be quoted as “a position paper of the European Renal Best Practice”

I the references 5-10 there are randomized controlled trials and one important was missed ( CREATE trial) thus the authors should modify the test accordingly

Material and Methods

Any information about iron supplement and iron parameters?. The authors just reported in the discussion section that “there were no differences in ferrokinetic, including transferrin saturation and serum ferritin concentrations, between the 2 groups after propensity score matching

What was the time of lab test? after the short or the long interval? Any information about the body weight increase at the time of the lab tests?

Statistics

Any information about the statistical power of the study? . Did you perform a multivariate analysis?

Results

Any information about CRP values?

Any different effect according to the underlined disease and dialysis technique ( HD or HDF)? (although the number of patients treated by HDF was rather small)

Limitations :the authors should acknowledged that 77 excluded patients because of lack of missing data is a significant number

Discussion

Pag. 10 : actually the ERBP suggest a target HB range of 10-12 gr/dL and the reference 5 is not an observational study but a randomized controlled trial: please modify the test accordingly

Reviewer #2: In this study, the authors hypothesize the existence of a turning point in the correlation of Hb and rSO2, with a stronger correlation at lower Hb levels. To study this, rSO2 is measured in 375 hemodialysis patients during 5 minutes before a HD session. The correlation between hemoglobin and rSO2 is studied for various consecutive thresholds. The turning point is chosen based on the highest slope and correlation coefficient in the lower Hb group. Next, propensity score matching is performed in a subgroup of n=168 and the correlation analysis is repeated. The authors conclude that a Hb of 10.8g/dL could be considered as a lower limit of a target Hb in order to prevent decrease of cerebral oxygenation. I have some comments on the study:

1. The hypothesis of a changing correlation at different Hb levels is interesting. However, I am doubtful about the statistical method. It seems an artificial approach to study subsequent thresholds until you find the highest correlation. Hemoglobin level is a continuous biological variable, and reducing it to a binary discards what may be important, useful data if you try to correlate it to rSO2. In reality the association between Hb and rSO2 might not change at one point, but be more a sliding scale. Second, cutpoints generally fit only to the currently observed data and are often not replicated in other independent studies or data sets. I would advise to perform a restricted cubic spline analysis. This will also allow to adjust for confounders in the model at the same time.

2. The authors suggest that a higher Hb (probably by increasing rSO2) would have positive effects on cognitive function. Indeed, the relationship between anemia and cognitive decline might be mediated by inadequate cerebral oxygen delivery and inadequate cerebral oxygenation. However, based on this observational study and other studies a causal effect cannot be stated. In other words, increasing Hb might increase rSO2. But whether a higher Hb, by increasing rSO2, will also improve cognitive function is unknown and questionable. Many other factors are playing a role in the cerebral oxygenation physiology, and Hb is only one factor. Similarly, many factors play a role in the complex pathophysiology of cognitive impairment and dementia, in which Hb is only one factor.

As an example: a study by Kovara et al (reference 15) in 39 HD patients showed no difference in Hb level between patients with normal versus impaired cognitive function.

3. The correlation between Hb and rSO2 is complex and other studies show opposite results. For example: a study in 18 HD patients (reference 12, Hoshin et al) showed no correlation between Hb (mean 10.4 g/dL) and rSO2. The Hb level is comparable to this study. Could the authors elaborate on this?

Reviewer #3: In this paper the Authors report the results of a multicenter observational study evaluating the relationship between Hb levels and brain oxygen saturation (rSO2), as assessed by near-infrared spectroscopy (NIRS), monitoring in 375 patients with end-stage kidney disease on routine hemodialysis (HD). The Authors explored this relationship by partitioning the patients above and below threshold Hb values in 0.1 g/dL increments across a range of 10.0 g/dL to 11.5 g/dL. They found that the slope of linear regression line and the value of the correlation coefficient between Hb and rSO2 were maximal below a 10.8 g/dL Hb value, and became shallower and non-significant, respectively, above this threshold. These results were consistent after propensity score matching of the patients for variables most impacting on rSO2. The Authors conclude that rSO2 is strongly dependent on Hb values below 10.8 g/dL, and that below this threshold lower Hb and rSO2 values may potentially have a negative impact on patients’ cognitive function.

The study is potentially interesting, as the combined effect of a suboptimal management of anemia and poor brain oxygenation may accelerate cognitive function decline in patients with ESKD on maintenance HD. However, several points need to be clarified:

1) The main limitation of this study, as also acknowledged by the Authors, is that it lacks clinical correlates and particularly measures of cognitive function. As such, the practical impact of their findings is unclear. This should be discussed in more details and stress further in the section dedicated to the limitation of the study.

2) The Authors do not mention before which session of the weekly HD routine the rSO2 measurements were performed. This is relevant, because a higher urea values, lower arterial pH and greater fluid overload at the end of the long interdialytic period may potentially impact on cerebral blood flow and arterial oxygen content.

3) In the same line of reasoning, did the Authors explore the impact on resting rSO2 of the interdialytic weight gain average in individual patients? Did a fraction of patients experience intradialytic hypotension? These issues are also relevant, because greater ultrafiltration volumes and/or intradialytic hypotension during the HD session preceding rSO2 measurement may impact on cerebral blood flow (see for example Regolisti G et al, Nephrol Dial Transplant 2013; 28:79-85).

4) The time period of patient enrollment should be specified.

5) The technique to perform propensity score matching is not mentioned. As the Authors used SPSS for statistical analyses, I assume that they either exploited the PS matching R software plug-in or the dedicated Python extension. Please explain.

6) Why was the relationship between Hb and rSO2 investigated across a narrower range of Hb values after (i.e., 10.5 g/dL to 11.0 g/dL) than before (10.0 g/dL to 11.5 g/dL) propensity score matching?

6. PLOS authors have the option to publish the peer review history of their article (what does this mean?). If published, this will include your full peer review and any attached files.

Reviewer #1: Yes: Prof. Francesco Locatelli

Reviewer #2: Yes: H.A. Polinder-Bos

Reviewer #3: No

---

## [Author Response · Author response to Decision Letter 0]

30 Jun 2020

Response to the editors’ and reviewers’ comments

We appreciate your careful review of our manuscript and thank you for the opportunity to submit our revised manuscript and the point-by-point responses to the academic editors’ and reviewers’ comments. We hope that we have satisfactorily addressed all issues raised by the academic editor and reviewers.

 

Academic editor

Comment 1:

The authors should clearly report ( both in the abstract and in the discussion) that if maintaining Hb levels above the lower limit would prevent the decrease of cerebral oxygenation, which might contribute to the maintenance of cognitive function, should be tested in a randomized controlled adequately powered trial specifically designed at this aim.

Response 1: 

Thank you for your insightful comment, and we completely agree with you. As advised by the reviewer and editor, we added these descriptions as a limitation in the “Discussion” section as follows:

Page 21, Line 12-Page 22, Line 7.

First, in this study, cognitive assessment could not be performed because of the limits of medical examination time for each patient. Renal anemia has been reported to be associated with cognitive impairment [32–34]. Thus, it would be expected that maintaining or improving cerebral rSO2 via the effect of the increased Hb levels is associated with changes in cognitive function. However, to date, there were only a few insufficient reports to examine the association between the cerebral oxygenation and cognitive function [16,17]. In addition, there was no difference in Hb levels between patients with normal versus cognitive impairment [16] and Hb levels showed no significant association with cerebral oxygenation in the multivariable linear regression analysis [17]. Consequently, the association between the cerebral rSO2 affected by the Hb levels and cognitive function remains uncertain. To confirm the effect of cerebral rSO2 affected by the Hb levels to the changes in cognitive function, a randomized controlled trial would be necessary.

In addition, we removed the sentences regarding the description between cerebral oxygenation and cognitive function in the Abstract of the revised manuscript.

Comment 2:

Please provide information about iron supplement and iron parameters

Response 2:

As advised by the reviewer and editor, we added the number of patients given iron supplementation, and data of serum iron concentration and total iron binding capacity, in addition to the transferrin saturation and serum ferritin concentration, in Table 1.

Comment 3:

Please specify before which session of the weekly HD routine the rSO2 measurements were performed

Response 3:

The number of patients at HD session who performed this study was distributed at first, second, and third HD sessions as follows:

Page 10, Lines 3-5.

In this study, the cerebral rSO2 measurement was performed at first HD session per weekly HD therapy in 241 patients, second HD session in 85 patients, and third HD session in 49 patients.

Comment 4:

Please consider performing additional statistical analyses as suggested by the Reviewers

Response 4:

In the revised manuscript, we performed a multivariable linear regression analysis to identify the factors associated with cerebral rSO2 values. Furthermore, the relationship between cerebral rSO2 and Hb levels was evaluated by a generalized linear model with restricted cubic spline function. As a result, Hb level was significantly associated with cerebral rSO2, and the two variables demonstrated a linear relationship.

Comment 5:

Please revise the suggested clinical implications

Response 5:

As mentioned above, the relationship between cerebral rSO2 and Hb level demonstrated a linear relationship. Therefore, the target Hb level in renal anemia management would be considered to be 11.5 g/dL, not exceeding 11.5 g/dL, or 12.0 g/dL, which does not exceed 12.0 g/dL, which were the upper limits of a target Hb level proposed for the appropriate management of renal anemia by previous guidelines and position statement. We added these descriptions in the revised manuscript as follows:

Page 20, Lines 6-10.

Therefore, from the viewpoint of maintaining cerebral oxygenation based on this study in HD patients, the target Hb level in renal anemia management would be considered to be 11.5 g/dL, not exceeding 11.5 g/dL, or 12.0 g/dL, not exceeding 12.0 g/dL, which were the upper limits of a target Hb level proposed for the appropriate management of renal anemia by previous guidelines and position statement [2-4].

Page 24, Lines 3-8.

Hb levels were significantly associated with cerebral rSO2 in the multivariable linear regression analysis. Furthermore, it was noted that cerebral rSO2 and Hb level had a linear relationship. Therefore, the target Hb level in renal anemia management would be considered to be the upper limits for the appropriate management of renal anemia by previous guidelines and position statement from the viewpoint of maintaining cerebral oxygenation in HD patients.

Comment 6:

Please acknowledge all the study limitation as pointed out by the Reviewers

Response 6:

As suggested by editor and reviewers, we corrected and added the description regarding the limitations of this study as follows:

Page 21, Line 12-Page 23, Line 10.

This study had several limitations which should be noted. First, in this study, cognitive assessment could not be performed because of the limits of medical examination time for each patient. Renal anemia has been reported to be associated with cognitive impairment [32–34]. Thus, it would be expected that maintaining or improving cerebral rSO2 via the effect of the increased Hb levels is associated with changes in cognitive function. However, to date, there were only a few insufficient reports to examine the association between the cerebral oxygenation and cognitive function [16,17]. In addition, there was no difference in Hb levels between patients with normal versus cognitive impairment [16] and Hb levels showed no significant association with cerebral oxygenation in the multivariable linear regression analysis [17]. Consequently, the association between the cerebral rSO2 affected by the Hb levels and cognitive function remains uncertain. To confirm the effect of cerebral rSO2 affected by the Hb levels to the changes in cognitive function, a randomized controlled trial would be necessary. Second, we could not unify the HD session of the week to evaluate cerebral oxygenation and blood samples in this study. Differences in interdialytic period in each patient may influence the results of blood chemistry and blood gas analysis as well as the increase in interdialytic weight gain; therefore, cerebral rSO2 values before HD might be influenced by interdialytic changes in these parameters. In this study, there were no significant associations between cerebral rSO2 before HD and interdialytic weight gain (kg), and interdialytic weight gain/dry weight (%) in a simple linear regression analysis, whereas we did not evaluate the cerebral rSO2 values at the end of the last HD session. Therefore, we cannot directly comment on the association between interdialytic changes in cerebral oxygenation and those in blood chemistry and blood gas analysis-associated parameters, and the increase in interdialytic weight gain. Third, inflammation shown in the elevated C-reactive protein is an exacerbating factor for renal anemia [35] and may be associated with changes in cerebral oxygenation via the decrease of Hb levels. However, in this study, not all patients included in this study measured C-reactive protein at the same time with cerebral rSO2 measurements. Therefore, we could not examine the influence of inflammation to the cerebral oxygenation. Furthermore, there were 77 excluded patients because of the insufficient application of blood samples in this study, and would be relatively large number to 452 included patients. Finally, this study could not determine the lower limit of a target Hb level because of the linear association between cerebral rSO2 and Hb levels. Therefore, additional studies are needed to confirm the association between Hb level and cerebral oxygenation in the future.

 

Reviewer 1

Comment 1:

Strictly speaking this was not an observational study considering that the authors planned to measure the cerebral oxygen saturation of patients undergoing hemodialysis.

Response 1: 

Thank you for your thoughtful comment. In the revised manuscript, we deleted the words “multi-center observational” because it could be misleading to the readers.

Comment 2:

The authors should clearly report ( both in the abstract and in the discussion session) that if maintaining Hb levels above the lower limit would prevent the decrease of cerebral oxygenation, which might contribute to the maintenance of cognitive function,. should be tested in a randomized controlled adequately powered trial specifically designed at this aim. The authors only reported that cerebral rSO2 showed a significant association with cognitive function in patients with advanced CKD, including those undergoing HD [15,16].Unfortunately association is not a demonstration.

Response 2:

Thank you for your insightful comment, and we completely agree with you. As advised by the reviewer, we added these descriptions as a limitation in the “Discussion” section as follows:

Page 21, Line 12-Page 22, Line 7.

First, in this study, cognitive assessment could not be performed because of the limits of medical examination time for each patient. Renal anemia has been reported to be associated with cognitive impairment [32–34]. Thus, it would be expected that maintaining or improving cerebral rSO2 via the effect of the increased Hb levels is associated with changes in cognitive function. However, to date, there were only a few insufficient reports to examine the association between the cerebral oxygenation and cognitive function [16,17]. In addition, there was no difference in Hb levels between patients with normal versus cognitive impairment [16] and Hb levels showed no significant association with cerebral oxygenation in the multivariable linear regression analysis [17]. Consequently, the association between the cerebral rSO2 affected by the Hb levels and cognitive function remains uncertain. To confirm the effect of cerebral rSO2 affected by the Hb levels to the changes in cognitive function, a randomized controlled trial would be necessary.

In addition, we removed the sentences regarding the description between cerebral oxygenation and cognitive function in the Abstract of the revised manuscript.

Comment 3:

The authors should clarify that only a severe anemia is responsible of the severe side effects reported here.

Response 3:

As advised by the reviewer, we added the description in the “Introduction” section as follows:

Page 5, Lines 4-5.

Under a severe anemic status caused by the progression of renal anemia,

Comment 4:

Reference 3 should be quoted as “a position paper of the European Renal Best Practice”

I the references 5-10 there are randomized controlled trials and one important was missed ( CREATE trial) thus the authors should modify the test accordingly

Response 4:

Thank you for your suggestion. We corrected the phrase from “the European Renal Association’s Best Practice Guideline” to “the European Renal Best Practice position statement”. Furthermore, we added the CREATE trial as reference no. 6 and corrected the words from “observational studies” to “randomized controlled trials and clinical trials” in the revised manuscript.

Comment 5:

Any information about iron supplement and iron parameters?. The authors just reported in the discussion section that “there were no differences in ferrokinetic, including transferrin saturation and serum ferritin concentrations, between the 2 groups after propensity score matching.

Response 5:

As advised by the reviewer, we added the number of patients given iron supplementation, and data of serum iron concentration and total iron binding capacity, in addition to the transferrin saturation and serum ferritin concentration, in Table 1. 

Comment 6:

What was the time of lab test? after the short or the long interval? Any information about the body weight increase at the time of the lab tests?

Response 6:

As pointed out by the reviewer, laboratory data would be influenced by the differences in interdialytic period. It would be very important to unify the HD session of weekly HD therapy for the evaluation in cerebral oxygenation and blood sample; however, unfortunately in this study, we could not unify the HD session of the week because we prioritized the cerebral oxygenation measurement and blood sample analysis in the same HD session. The number of patients in a HD session who were a part of this study was distributed at first, second, and third HD sessions as follows:

Page 10, Lines 3-5.

In this study, the cerebral rSO2 measurement was performed at first HD session per weekly HD therapy in 241 patients, second HD session in 85 patients, and third HD session in 49 patients.

Furthermore as advised by the reviewer, we confirmed the association between cerebral rSO2 values before HD and interdialytic weight gain from the end of the last HD session to the current HD session. In this study, interdialytic weight gain was 1.9 ± 1.0 kg and interdialytic weight gain/dry weight was 3.4 ± 1.8%. No significant associations were confirmed between cerebral rSO2 and interdialytic weight gain (r = 0.04, p = 0.49) and cerebral rSO2 and interdialytic weight gain/dry weight (r = 0.07, p = 0.19). We added these results in Table 1.

Comment 7:

Any information about the statistical power of the study?

Response 7:

In the revised manuscript, we changed the method to evaluate the association using a generalized linear model with restricted cubic spline function between cerebral rSO2 and Hb levels. The comparison between these two variables was not performed; therefore, we did not calculate the statistical power in the revised manuscript.

Comment 8:

Did you perform a multivariate analysis?

Response 8:

Yes, we performed the multivariable linear regression analysis in the revised manuscript. Variables that were significantly correlated with cerebral rSO2 in a simple linear regression analysis were included in the multivariable linear regression analysis to identify factors affecting cerebral rSO2 in HD patients. Throughout these analyses, Hb levels were significantly associated with cerebral rSO2 values. We added these descriptions in the “Results” section and Table 2 as follows:

Page 15, Line 9-Page 16, Line 15.

Results of the multivariable linear regression analysis are presented in Table 2. Ln-HD duration, mean blood pressure, dose of ESA, pH, Hb level, transferrin saturation, serum creatinine, and serum albumin as variables with p values below 0.05 and presence of DM as a confounding factor were included in the multivariable linear regression analysis. As shown in Table 2, cerebral rSO2 was independently associated with Ln-HD duration (standardized coefficient: -0.36), mean blood pressure (standardized coefficient: 0.13), pH (standardized coefficient: -0.10), serum albumin (standardized coefficient: 0.14), presence of DM (standardized coefficient: -0.20), and Hb level (standardized coefficient: 0.29).

Table 2. Multivariable linear regression analysis: independent factors of cerebral rSO2 in HD patients

.

Variables Coefficient Standardized coefficient p

Ln-HD duration -1.71 -0.36 < 0.001*

Mean blood pressure 0.08 0.13 0.005*

Erythropoiesis-stimulating agent -0.04 0.437

pH -22.9 -0.10 0.024*

Hb 1.83 0.29 < 0.001*

Serum creatinine 0.05 0.360

Serum albumin 2.40 0.14 0.007*

Transferrin saturation 0.06 0.144

Presence of diabetes mellitus -3.89 -0.20 < 0.001*

*Statistically significant.

Abbreviations: HD, hemodialysis; Hb, hemoglobin

Comment 9:

Any information about CRP values?

Response 9:

As pointed out by the reviewer, it is very important to analyze the associations among cerebral oxygenation, Hb level, and inflammation including C-reactive protein in the clinical setting of HD therapy. Unfortunately, C-reactive protein was not measured in all patients at the same time when cerebral rSO2 was measured in this study. Therefore, C-reactive protein was excluded in this study. We will confirm the association among cerebral oxygenation, Hb level, and C-reactive protein in HD patients in the future. We added these descriptions to the limitation in the “Discussion” section as follows:

Page 22, Line 18-Page 23, Line 5.

Third, inflammation shown in the elevated C-reactive protein is an exacerbating factor for renal anemia [35] and may be associated with changes in cerebral oxygenation via the decrease of Hb levels. However, in this study, not all patients included in this study measured C-reactive protein at the same time with cerebral rSO2 measurements. Therefore, we could not examine the influence of inflammation to the cerebral oxygenation.

Comment 10:

Any different effect according to the underlined disease and dialysis technique ( HD or HDF)? (although the number of patients treated by HDF was rather small)

Response 10:

As advised by the reviewer, cerebral rSO2 values were compared between HD (n = 310) and HDF patients (n = 65). Cerebral rSO2 values in HD patients and those in HDF were 51.5 ± 9.5% and 48.7 ± 9.4%, respectively. There was no significant difference between the two groups (p = 0.06) in this study. Presence of DM was reported to be significantly and negatively associated with cerebral rSO2 in HD patients (Reference no 14). As same as previous report, cerebral rSO2 values in HD patients with DM were significantly lower than those in HD patients with other diseases using one-way analysis of variance (DM patients, 48.6 ± 9.3%; chronic glomerulonephritis patients, 53.2 ± 9.9%; nephrosclerosis patients, 53.6 ± 8.0%; others, 52.4 ± 9.7%, p <0.001 vs. other diseases) in this study. Therefore, presence of DM was included in the multivariable linear regression analysis to identify factors affecting cerebral rSO2, and in a generalized linear model with restricted cubic spline function in this study. We added these descriptions in the “Results” section in the revised manuscript as follows:

Page 12, Lines 10-12.

Furthermore, cerebral rSO2 was significantly lower in HD patients with DM than in those without DM (48.6 ± 9.3% vs. 52.9 ± 9.3%, p < 0.001, Fig 2).

Page 15, Lines 9-17.

Results of the multivariable linear regression analysis are presented in Table 2. Ln-HD duration, mean blood pressure, dose of ESA, pH, Hb level, transferrin saturation, serum creatinine, and serum albumin as variables with p values below 0.05 and presence of DM as a confounding factor were included in the multivariable linear regression analysis. As shown in Table 2, cerebral rSO2 was independently associated with Ln-HD duration (standardized coefficient: -0.36), mean blood pressure (standardized coefficient: 0.13), pH (standardized coefficient: -0.10), serum albumin (standardized coefficient: 0.14), presence of DM (standardized coefficient: -0.20), and Hb level (standardized coefficient: 0.29).

Page 16, Line 17-Page 17, Line 5.

Furthermore, we conducted a regression analysis using a generalized linear model with restricted cubic spline function to test a non-linear relationship between cerebral rSO2 and Hb levels. We adjusted for Ln-HD duration, mean blood pressure, pH, serum albumin, and presence of DM, which were independently and significantly associated with cerebral rSO2 in the multivariable linear regression analysis, and the cerebral rSO2 and Hb levels demonstrated a linear relationship (p for linearity = 0.79, Fig 3). 

Comment 11:

Limitations :the authors should acknowledged that 77 excluded patients because of lack of missing data is a significant number.

Response 11:

Thank you for pointing out this important aspect. We understand that we excluded 77 patients in this study; we shall acknowledge the same in limitations. Applications of the blood sample examination, including peripheral blood count, blood chemistry, and blood gas analysis, were performed manually. Even though skilled medical staff was readily accessible, these tests may have been challenging to accurately perform for unskilled medical staff. As a result, we reported significant lack of data with respect to these 77 patients, including peripheral blood count, blood chemistry, and blood gas analysis, in this study, and unfortunately, had to exclude them. We added these descriptions to the limitation in the “Discussion” section as follows:

Page 23, Lines 5-7.

Furthermore, there were 77 excluded patients because of the insufficient application of blood samples in this study, and would be relatively large number to 452 included patients.

Comment 12:

Pag. 10 : actually the ERBP suggest a target HB range of 10-12 gr/dL and the reference 5 is not an observational study but a randomized controlled trial: please modify the test accordingly.

Response 12:

As advised by the reviewer, we corrected the sentence to “The European Renal Best Practice position statement recommended that Hb values should not be allowed to routinely fall below 10 g/dL and that the start of ESA therapy could be considered at higher Hb values but not exceeding 12 g/dL”. Furthermore, we changed the sentence to “based on the results of randomized controlled trials and clinical trials [5-11].”

 

Reviewer 2

Comment 1:

The hypothesis of a changing correlation at different Hb levels is interesting. However, I am doubtful about the statistical method. It seems an artificial approach to study subsequent thresholds until you find the highest correlation. Hemoglobin level is a continuous biological variable, and reducing it to a binary discards what may be important, useful data if you try to correlate it to rSO2. In reality the association between Hb and rSO2 might not change at one point, but be more a sliding scale.

Response 1: 

Thank you for your important suggestion, and we completely agree with your comments. In the revised manuscript, we completely changed the method to evaluate the association using a generalized linear model with restricted cubic spline function between cerebral rSO2 and Hb levels.

Comment 2:

Second, cutpoints generally fit only to the currently observed data and are often not replicated in other independent studies or data sets. I would advise to perform a restricted cubic spline analysis. This will also allow to adjust for confounders in the model at the same time.

Response 2:

Thank you for your important suggestion, and we agree with your comments. In the revised manuscript, we performed a generalized linear model with restricted cubic spline function. As a result, cerebral rSO2 and Hb level was proven to have a linear relationship (p for linearity = 0.79) after the adjustment with other factors significantly associated with cerebral rSO2. Therefore, from the viewpoint of maintaining cerebral oxygenation based on this study in HD patients, the target Hb level in renal anemia management would be considered to be 11.5 g/dL, not exceeding 11.5 g/dL, or 12.0 g/dL, which were the upper limits of the target Hb level proposed for the appropriate management of renal anemia by previous guidelines and position statement.

Comment 3:

The authors suggest that a higher Hb (probably by increasing rSO2) would have positive effects on cognitive function. Indeed, the relationship between anemia and cognitive decline might be mediated by inadequate cerebral oxygen delivery and inadequate cerebral oxygenation. However, based on this observational study and other studies a causal effect cannot be stated. In other words, increasing Hb might increase rSO2. But whether a higher Hb, by increasing rSO2, will also improve cognitive function is unknown and questionable. Many other factors are playing a role in the cerebral oxygenation physiology, and Hb is only one factor. Similarly, many factors play a role in the complex pathophysiology of cognitive impairment and dementia, in which Hb is only one factor. As an example: a study by Kovara et al (reference 15) in 39 HD patients showed no difference in Hb level between patients with normal versus impaired cognitive function.

Response 3:

We agree with your comment. As advised by the reviewer, we added these descriptions to the limitation in the “Discussion” section as follows:

Page 21, Line 12-Page 22, Line 7.

First, in this study, cognitive assessment could not be performed because of the limits of medical examination time for each patient. Renal anemia has been reported to be associated with cognitive impairment [32–34]. Thus, it would be expected that maintaining or improving cerebral rSO2 via the effect of the increased Hb levels is associated with changes in cognitive function. However, to date, there were only a few insufficient reports to examine the association between the cerebral oxygenation and cognitive function [16,17]. In addition, there was no difference in Hb levels between patients with normal versus cognitive impairment [16] and Hb levels showed no significant association with cerebral oxygenation in the multivariable linear regression analysis [17]. Consequently, the association between the cerebral rSO2 affected by the Hb levels and cognitive function remains uncertain. To confirm the effect of cerebral rSO2 affected by the Hb levels to the changes in cognitive function, a randomized controlled trial would be necessary.

Comment 4:

The correlation between Hb and rSO2 is complex and other studies show opposite results. For example: a study in 18 HD patients (reference 12, Hoshin et al) showed no correlation between Hb (mean 10.4 g/dL) and rSO2. The Hb level is comparable to this study. Could the authors elaborate on this?

Response 4:

In this study, we collected large number of HD patients with widely distributed Hb levels compared with previous studies [Hoshino et al, Ito K et al], and investigated to identify the clinical parameters significantly associated cerebral rSO2 values before HD. In the multivariable linear regression analysis, Hb levels were significantly associated with cerebral rSO2, different from previous reports. The causes of the differences of influence in Hb levels to the cerebral oxygenation may owe to the differences in the number of included patients and Hb level distribution. We added these descriptions in the “Discussion” section as follows:

Page 19, Lines 2-9.

Therefore, we collected large number of HD patients with widely distributed Hb levels compared with previous studies [13,14], and investigated to identify the clinical parameters, including Hb level, significantly associated with cerebral rSO2 values. As a result, cerebral rSO2 was significantly associated with Hb levels in addition to the HD duration, mean blood pressure, pH, serum albumin, and presence of DM. Furthermore, after the adjustment for Ln-HD duration, mean blood pressure, pH, serum albumin, and presence of DM using multivariable analysis, linear association between cerebral rSO2 and Hb levels was proven in this study.

 

Reviewer 3

Comment 1:

The main limitation of this study, as also acknowledged by the Authors, is that it lacks clinical correlates and particularly measures of cognitive function. As such, the practical impact of their findings is unclear. This should be discussed in more details and stress further in the section dedicated to the limitation of the study.

Response 1: 

As advised by the reviewer, we added the description regarding the uncertain association between cerebral oxygenation influenced by Hb levels and cognitive function to the limitation in the “Discussion” section in the revised manuscript.

Page 21, Line 12-Page 22, Line 7.

First, in this study, cognitive assessment could not be performed because of the limits of medical examination time for each patient. Renal anemia has been reported to be associated with cognitive impairment [32–34]. Thus, it would be expected that maintaining or improving cerebral rSO2 via the effect of the increased Hb levels is associated with changes in cognitive function. However, to date, there were only a few insufficient reports to examine the association between the cerebral oxygenation and cognitive function [16,17]. In addition, there was no difference in Hb levels between patients with normal versus cognitive impairment [16] and Hb levels showed no significant association with cerebral oxygenation in the multivariable linear regression analysis [17]. Consequently, the association between the cerebral rSO2 affected by the Hb levels and cognitive function remains uncertain. To confirm the effect of cerebral rSO2 affected by the Hb levels to the changes in cognitive function, a randomized controlled trial would be necessary.

Comment 2:

The Authors do not mention before which session of the weekly HD routine the rSO2 measurements were performed. This is relevant, because a higher urea values, lower arterial pH and greater fluid overload at the end of the long interdialytic period may potentially impact on cerebral blood flow and arterial oxygen content.

Response 2: 

As pointed out by the reviewer, it would be very important to unify the HD session of weekly HD therapy for the cerebral oxygenation and blood sample evaluation in each patient. However, in this study, we could not unify the HD session of the week because we prioritized cerebral oxygenation measurement and blood sample analysis on the same HD session. As suggested, changes in the blood chemistry and blood gas analysis parameters, and the increase in interdialytic weight gain at the end of the long interdialytic period may influence the cerebral oxygenation before HD. In this study, interdialytic weight gain was 1.9 ± 1.0 kg and interdialytic weight gain/dry weight was 3.4 ± 1.8%. No significant associations were confirmed between cerebral rSO2 and interdialytic weight gain (r = 0.04, p = 0.49) and cerebral rSO2 and interdialytic weight gain / dry weight (r = 0.07, p = 0.19). However, we did not evaluate the cerebral rSO2 values at the end of the last HD session. Therefore, we cannot directly comment on the association between the changes in cerebral oxygenation and those in blood chemistry and blood gas analysis-associated parameters as well as the increase in interdialytic weight gain. We added these descriptions as a limitation in the “Discussion” section. Furthermore, the number of patients at HD session who performed this study was distributed at first, second, and third HD sessions as follows:

Page 22, Line 7-Page 22, Line 18.

Second, we could not unify the HD session of the week to evaluate cerebral oxygenation and blood samples in this study. Differences in interdialytic period in each patient may influence the results of blood chemistry and blood gas analysis as well as the increase in interdialytic weight gain; therefore, cerebral rSO2 values before HD might be influenced by interdialytic changes in these parameters. In this study, there were no significant associations between cerebral rSO2 before HD and interdialytic weight gain (kg), and interdialytic weight gain/dry weight (%) in a simple linear regression analysis, whereas we did not evaluate the cerebral rSO2 values at the end of the last HD session. Therefore, we cannot directly comment on the association between interdialytic changes in cerebral oxygenation and those in blood chemistry and blood gas analysis-associated parameters, and the increase in interdialytic weight gain.

Page 10, Lines 3-5.

In this study, the cerebral rSO2 measurement was performed at first HD session per weekly HD therapy in 241 patients, second HD session in 85 patients, and third HD session in 49 patients.

Comment 3:

In the same line of reasoning, did the Authors explore the impact on resting rSO2 of the interdialytic weight gain average in individual patients? Did a fraction of patients experience intradialytic hypotension? These issues are also relevant, because greater ultrafiltration volumes and/or intradialytic hypotension during the HD session preceding rSO2 measurement may impact on cerebral blood flow (see for example Regolisti G et al, Nephrol Dial Transplant 2013; 28:79-85).

Response 3: 

Thank you for your thoughtful comment. As advised by the reviewer, we confirmed the association between cerebral rSO2 values before HD and interdialytic weight gain from the end of the last HD session to the current HD session. In this study, interdialytic weight gain was 1.9 ± 1.0 kg and interdialytic weight gain / dry weight was 3.4 ± 1.8%. No significant associations were confirmed between cerebral rSO2 and interdialytic weight gain (r = 0.04, p = 0.49) and cerebral rSO2 and interdialytic weight gain / dry weight (r = 0.07, p = 0.19). However, the influence of interdialytic weight gain to the changes in cerebral oxygenation could not be completely excluded in this study; therefore, we added the description regarding the possibility that interdialytic weight gain may influence the interdialytic changes in cerebral oxygenation to the limitation in the “Discussion” section as follows:

Page 22, Line 7-Page 22, Line 18.

Second, we could not unify the HD session of the week to evaluate cerebral oxygenation and blood samples in this study. Differences in interdialytic period in each patient may influence the results of blood chemistry and blood gas analysis as well as the increase in interdialytic weight gain; therefore, cerebral rSO2 values before HD might be influenced by interdialytic changes in these parameters. In this study, there were no significant associations between cerebral rSO2 before HD and interdialytic weight gain (kg), and interdialytic weight gain/dry weight (%) in a simple linear regression analysis, whereas we did not evaluate the cerebral rSO2 values at the end of the last HD session. Therefore, we cannot directly comment on the association between interdialytic changes in cerebral oxygenation and those in blood chemistry and blood gas analysis-associated parameters, and the increase in interdialytic weight gain.

As pointed out by the reviewer, the changes in blood volume and/or those in blood pressure during HD would influence the cerebral blood flow and oxygen supply to the brain [Regolisti G, et al. Nephrol Dial Transplant. 2013;28:79-85, Polinder-Bos HA, et al. J Am Soc Nephrol. 2018;29:1317-1325]. However, in this study, cerebral rSO2 measurements were performed only before HD; therefore, we could not evaluate the association between the changes in blood volume and/or blood pressure and those in cerebral rSO2 during HD. In addition, HD patients with symptomatic intradialytic hypotension were not included in this study. We added these description in the “Discussion” section as follows:

Page 20, Line 11-Page 21, Line 11.

In this study, mean blood pressure showed positive and significant association with cerebral rSO2 in the multivariable linear regression analysis, which was different from the previous report [14]. Cerebral blood flow is held steady between mean blood pressure 60 and 150 mmHg, which was known as cerebral autoregulation to protect the cerebral tissue from fluctuating systemic blood pressure [28]. Furthermore, cerebral rSO2 was reported to become linearly dependent on BP, in particular, below a mean blood pressure of 60 mmHg [29]. Additionally, in HD patients, impaired cerebral autoregulation due to vascular damages and HD-associated risk factors for hemodynamic instability during HD are likely to lead to the cerebral hypoperfusion [30]. Actually, cerebral perfusion pressure, defined as the difference between mean blood pressure and intracranial pressure, depends largely on the mean blood pressure during HD, and a larger ultrafiltration volume and rate, which sometimes lead to the decrease in mean blood pressure, were reported to be associated with lower cerebral blood flow [31]. Based on these reports, mean blood pressure might be associated with changes in oxygen supply and those in cerebral oxygenation through the impaired cerebral autoregulation and hemodynamic stress during HD. However, in this study, cerebral rSO2 measurements were performed only before HD. Therefore, we cannot directly comment on the association between changes in cerebral oxygenation and those in mean blood pressure influenced by the hemodynamic stress during HD.

Comment 4:

The time period of patient enrollment should be specified.

Response 4: 

As suggested, we added the description regarding the enrollment period in the “Materials and Methods” section as follows:

Page 7, Lines 10-11.

A total of 452 patients met the inclusion criteria and were enrolled from July 1, 2013 to August 31, 2017.

Comment 5:

The technique to perform propensity score matching is not mentioned. As the Authors used SPSS for statistical analyses, I assume that they either exploited the PS matching R software plug-in or the dedicated Python extension. Please explain.

Response 5: 

Thank you for your comment. In the revised manuscript, we changed the statistical method to confirm the association between Hb level and cerebral oxygenation. Therefore, we did not perform propensity score matching in the revised manuscript. We rewrote “Statistics” section as follows:

Page 10, Line 13-Page 11, Line 7.

Data were expressed as mean ± standard deviation or median and interquartile range. Correlations between the two groups were evaluated using the Pearson’s correlation coefficient and linear regression analysis. The Student’s t-test for non-paired values was used for comparing 2 groups. Variables with a p value below 0.05 in a simple linear regression analysis and plausible confounding factors were included in the multivariable linear regression analysis to identify factors affecting cerebral rSO2 in HD patients. Furthermore, a generalized linear model with restricted cubic spline function was used to investigate the non-linear association between cerebral rSO2 and Hb levels. HD duration was transformed using the natural logarithm (Ln) in the regression analyses because of the skewed distribution. All analyses were performed using the IBM SPSS Statistics for Windows, version 26.0 (IBM, Armonk, NY, USA). Additionally, R V.3.4.1 (The R Foundation, Vienna, Austria) and ‘rms’ package was used for a restricted cubic spline function. P < 0.05 was considered statistically significant.

Comment 6:

Why was the relationship between Hb and rSO2 investigated across a narrower range of Hb values after (i.e., 10.5 g/dL to 11.0 g/dL) than before (10.0 g/dL to 11.5 g/dL) propensity score matching?

Response 6: 

Thank you for your comment. In the revised manuscript, we changed the analysis method to confirm the association between Hb level and cerebral oxygenation. Therefore, we removed this Table in the revised manuscript.

We hope that our revised manuscript is now suitable for publication in your highly esteemed journal.

---

## [Decision Letter · Decision Letter 1]

14 Jul 2020

Associations of cerebral oxygenation with hemoglobin levels evaluated by near-infrared spectroscopy in hemodialysis patients

PONE-D-20-05681R1

Dear Dr. Ookawara,

We’re pleased to inform you that your manuscript has been judged scientifically suitable for publication and will be formally accepted for publication once it meets all outstanding technical requirements.

Kind regards,

Justyna Gołębiewska

Academic Editor

PLOS ONE

Additional Editor Comments (optional):

Reviewers' comments:

Reviewer's Responses to Questions

**Comments to the Author**

1. If the authors have adequately addressed your comments raised in a previous round of review and you feel that this manuscript is now acceptable for publication, you may indicate that here to bypass the “Comments to the Author” section, enter your conflict of interest statement in the “Confidential to Editor” section, and submit your "Accept" recommendation.

Reviewer #1: All comments have been addressed

Reviewer #2: All comments have been addressed

Reviewer #3: All comments have been addressed

2. Is the manuscript technically sound, and do the data support the conclusions?

Reviewer #1: Yes

Reviewer #2: Yes

Reviewer #3: Yes

3. Has the statistical analysis been performed appropriately and rigorously? 

Reviewer #1: Yes

Reviewer #2: Yes

Reviewer #3: Yes

4. Have the authors made all data underlying the findings in their manuscript fully available?

Reviewer #1: Yes

Reviewer #2: Yes

Reviewer #3: (No Response)

5. Is the manuscript presented in an intelligible fashion and written in standard English?

Reviewer #1: Yes

Reviewer #2: Yes

Reviewer #3: Yes

6. Review Comments to the Author

Reviewer #1: The authors sorted out all my criticisms thus no furher requests from my side condisering that the authors acknowledged the limitations of the present study

Reviewer #2: The authors have substantially improved the manuscript. Only 2 minor comments:

1. The authors write in the Discussion about ' impaired cerebral autoregulation due to vascular damages'. In literature several authors write about impaired cerebral auto regulation in dialysis patients, however this is a hypothesis that has never been studied or proven by my knowledge..

2. Some typo's in the references: reference 30: 'Wolfgram' instead of ' Wolfgam'.

Reference 31: 'Polinder' instead of ' Pollinder'.

Reviewer #3: The Authors have exhaustively and appropriately addressed all of the criticisms I raised in my review.

7. PLOS authors have the option to publish the peer review history of their article (what does this mean?). If published, this will include your full peer review and any attached files.

Reviewer #1: **Yes: **Francesco Locatelli

Reviewer #2: **Yes: **Harmke Polinder-Bos, MD PhD

Reviewer #3: No